# Chiral electrocatalysts eclipse water splitting metrics through spin control

Aravind Vadakkayil [1], Caleb Clever[1], Karli N. Kunzler[1], Susheng Tan [2,3], Brian P. Bloom [1] ✉ & David H. Waldeck [1,2] ✉

Continual progress in technologies that rely on water splitting are often hampered by the slow kinetics associated with the oxygen evolution reaction (OER). Here, we show that the efficiency of top-performing catalysts can be improved, beyond typical thermodynamic considerations, through control over reaction intermediate spin alignment during electrolysis. Spin alignment is achieved using the chiral induced spin selectivity (CISS) effect and the improvement in OER manifests as an increase in Faradaic efficiency, decrease in reaction overpotential, and change in the rate determining step for chiral nanocatalysts over compositionally analogous achiral nanocatalysts. These studies illustrate that a defined spatial orientation of the nanocatalysts is not necessary to exhibit spin selectivity and therefore represent a viable platform for employing the transformative role of chirality in other reaction pathways and processes.

The oxygen evolution reaction (OER) remains a significant bottleneck for numerous electrocatalytic and electrochemical processes, including water electrolysis[1,2], the electrochemical reduction of $CO_2$[3,4], and exchange membranes for batteries and fuel cells[5–7], among others. The creation of molecular oxygen by anodic electrocatalysts is believed to proceed on their surface through a number of possible radical intermediates, adsorbed *OH, *O, and *OOH[8]. The overall reaction is a four-electron, four-proton process and a number of mechanistic schemes have been proposed[9]. Because of the overarching complexity involved in the OER, significant work has focused on developing simplified models to guide catalyst design. For instance, the catalytic activity depends on the adsorption binding energy differences of the reactive intermediates on the surface: binding too strongly can hinder the progression of the reaction whereas not binding strongly enough leads to desorption before the reaction can proceed[10]. A plot of the dependence of reaction rate (or overpotential) on the binding energy changes for different electrocatalysts manifests as a volcano plot, in which the apex represents the best catalyst characteristics. Similar volcano plots have also been made for mixed metal oxide catalysts using the occupancy of the $e_g$ orbitals to define the optimal performance[11]. While strategies such as these have resulted in

significant progress in advancing catalyst design, these models often ignore the role of the electron's spin on the reaction kinetics. Because the ground state of diatomic oxygen exists as a triplet, spin constraints should affect the elementary reaction steps[12–15].

By spin filtering the anodic current, one can create spin-polarized reaction intermediates during electrocatalysis, and hence improve the efficiency of the OER. For instance, an external magnetic field with a ferromagnetic electrode[16] can spin filter electron currents. Garces-Pineda and coworkers showed that a significant increase in current density (circa 100 mA/cm²) during the OER occurs for magnetized ferromagnetic mixed metal oxide catalysts compared to their non-magnetized counterparts[17]. In related work, Ren et al. showed that applying an external magnetic field during OER changed the rate-determining step[18]. Alternatively one can use the chiral-induced spin selectivity (CISS) effect [19–22] with chiral electrocatalysts to spin filter the anodic current. In proof-of-principle experiments, we and others have shown that CISS improves both the Faradaic efficiency and reaction overpotential, $\eta$, of the OER[12,14,23–26]. Deviations in standard volcano plots for mixed metal oxide catalysts have been proposed[27], and some evidence for chiral $NiO_x$ catalysts exceeding volcano plot restrictions has been reported[28]. The electrocatalysts' performance in

[1]Chemistry Department, University of Pittsburgh, Pittsburgh, PA 15260, USA. [2]Petersen Institute of Nanoscience and Engineering, University of Pittsburgh, Pittsburgh, PA 15260, USA. [3]Department of Electrical and Computer Engineering, University of Pittsburgh, Pittsburgh, PA 15260, USA. ✉e-mail: bpb8@pitt.edu; dave@pitt.edu

these proof-of-principle studies, as well as those using magnetic electrodes, does not compete with benchmark catalysts, however. Moreover, the ability to scale the favorable spin effects to industrially-relevant applications is unexplored. This work presents a scalable method for the creation of chiral cobalt oxide nanoparticle catalysts doped with iron ($Co_{(3-x)}Fe_xO_4$) that display a superior catalytic performance than that of achiral $Co_{(3-x)}Fe_xO_4$ and of $IrO_2$ in alkaline media.

## Results and discussion

### Synthesis and morphology

Chiral $Co_{(3-x)}Fe_xO_4$ catalysts were synthesized by adapting published protocols with cysteine as a capping ligand[29]. To compare the catalytic activity to an achiral analog, a racemic mixture of cysteine (Rac) was used. Iron-doped nanoparticles were synthesized through the addition of iron (III) chloride during synthesis and the amount of dopant was kept the same for L- and Rac-cobalt oxide (see Methods and Supplementary Table S1). Figure 1a, b show absorbance and corresponding circular dichroism spectra for undoped (black), 5% Fe (green), 10% Fe (blue), and 23% Fe (purple) during the synthesis of Rac- (light, dashed) and L-cobalt oxide (dark, solid) nanoparticles. The features at 230, 280, and 350 nm have previously been attributed to Co(II) → Co(III) intraparticle transitions and the features at 450, 550, and 600 nm to surface state (including ligand) → Co(III) transitions[29]. The broadened spectral features upon doping likely reflect the change in relative cobalt content per nanoparticle, the amount of surface defects in the particles, as well as new features arising from the iron dopant. Such structural and electronic changes to cobalt oxide upon iron doping have been reported previously[30]. Despite these differences, the magnitude of the chiroptical dissymmetry in the CD spectra is similar and affirms strong chiral imprinting onto the catalyst's density-of-states. Figure 1c, d show scanning transmission electron microscopy (STEM) images for the eight different catalyst materials, all of which possess an average size between 4 and 5 nm.

### Catalytic activity and performance metrics

To examine the OER activity of the different electrocatalysts, an ink suspension comprising the catalyst nanoparticles in a Nafion binder was made, using standard protocols[31], and dropcast onto a glassy carbon working electrode (see Methods for a detailed procedure). Figure 2a–d shows the *iR*-corrected linear sweep voltammograms (LSV) collected for all of the different catalysts at a scan rate of 10 mV/s in a 1 M NaOH electrolyte solution. Note, the current is normalized to the geometric area and the Nafion binder's contribution to the current was subtracted. The LSVs show a large decrease in the reaction overpotential, $\eta$, for the chiral cobalt oxide catalyst (black, solid line) over the achiral equivalent (gray, dashed line) and this behavior persists across all of the iron dopant concentrations: 5% (green), 10% (blue), and 23% (purple). LSV's that are normalized to the electrochemical surface area (ECSA), see representative measurement in Supplementary Fig. S1, are shown in Supplementary Fig. S2 for comparison and indicate that the change in $\eta$ is not a surface area related phenomenon. Figure 2e reports the $\eta$ at 10 mA cm$^{-2}$ for all of the catalysts and Supplementary Fig. S3 and Table S2 report stability test data following 2 h of constant operation at 10 mA cm$^{-2}$. The addition of iron dopants to the cobalt oxide reduces the $\eta$ for the OER in a manner similar to that reported in previous studies[32]. Interestingly, the decrease in $\eta$ for chiral catalysts over their achiral analogs, Fig. 2e, is approximately the same among all of the different Fe percent dopants; e.g., the curves of the

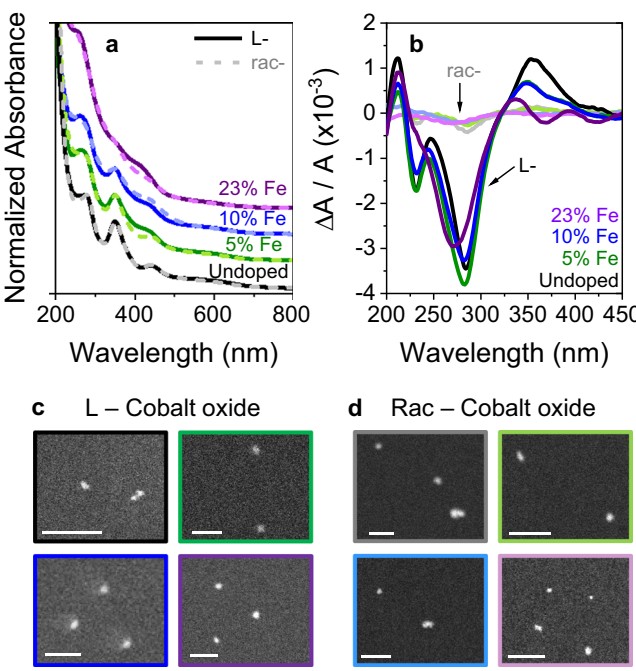

**Fig. 1 | Characterization of catalysts.** Absorbance (**a**) and circular dichroism (**b**) spectra of undoped (black) and 5% (green), 10% (blue), and 23% (purple) iron-doped Rac- (light, dashed line) and L-cobalt oxide (dark, solid line) nanoparticles. Note that the spectra in (**a**) are displaced from each other for clarity. **c**, **d** Show representative STEM images of the catalysts for L-cobalt oxide and Rac-cobalt oxide respectively. The color of the border reflects the same color coding as used in (**a**, **b**) and the scale bar in the images is 25 nm.

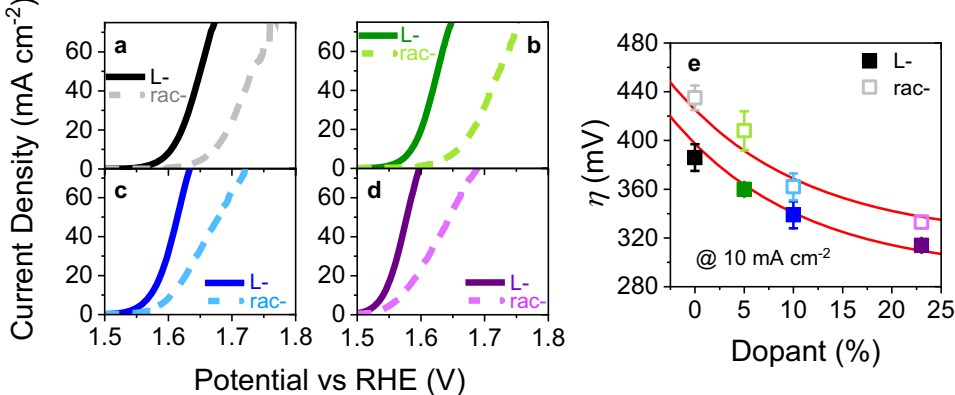

**Fig. 2 | Voltammetric properties. a–d** Shows linear sweep voltammograms of undoped (black) and 5% (green), 10% (blue), and 23% (purple) iron-doped Rac- (dashed line) and L-cobalt oxide (solid line) nanoparticle catalysts in Nafion measured in a 1 M NaOH electrolyte. **e** Shows the corresponding overpotential at 10 mA cm-2 for Rac- (open symbols) and L-cobalt oxide (filled symbols). The red lines are a visual guide for emphasizing the effect of chirality and the error bars represent the standard deviation from at least three independent measurements.

two red lines as drawn are equivalent and offset by 28 mV. Note that $Co_3O_4$ is close to the apex in volcano plots[10]; it provides a nearly optimum set of adsorbate stabilities for the reaction intermediates. The systematic deviation of the overpotentials for the different compositions implies that chirality is a wholly different design variable that can be used to reduce $\eta$ in the OER.

Because the overpotential is affected by the catalyst loading, catalyst geometry, and the electrode configuration (e.g., if it is deposited on glassy carbon, nickel foam, carbon cloth, etc.)[33], we report the mass activity, MA, in which the current is normalized to the mass of the added catalyst, and the specific activity, SA, in which the current is normalized to the electrochemical surface area, see Fig. 3a, b and Supplementary Table S2. A greater than twofold enhancement in MA and SA, calculated at a 350 mV overpotential, is found for the chiral catalysts (dark, filled color) over their achiral analogs (light, open-color). Moreover, the 23% Fe-doped chiral cobalt oxide catalysts possess a >400-fold higher MA ($1730 \pm 178$ A g$^{-1}$) and >200-fold higher SA ($1.18 \pm 0.11$ mA cm$^{-2}$) compared to benchmark $IrO_2$ catalysts (4.2 A g$^{-1}$, 0.005 mA cm$^{-2}$) under similar electrolyte conditions[34]. To the best of our knowledge, the MAs are comparable to or exceed some of the highest reported values in the literature when accounting for the total catalyst mass, i.e., not just the metal loading, at $\eta = 350$ mV[35,36]. See Supplementary Table S2 for details regarding the different catalysts studied in this work, as well as comparisons to other relevant materials. Because these metrics are associated with the total current at a given potential, and not directly with the amount of $O_2$ production, rotating

ring-disk electrode measurements were employed to measure the Faradaic efficiency of the reaction; see Methods for details regarding the experiments.

Figure 3c shows the ratio of the Faradaic efficiency, FE$_L$, for undoped L-cobalt oxide (black) and 23% Fe-doped L-cobalt oxide catalysts (purple), to that of Rac-cobalt oxide catalysts, FE$_{Rac}$, in 1 M NaOH (horizontal dash) and in 0.1 M pH 10 (dotted) and 0.02M pH 8 (cross-hatched) sodium carbonate and potassium phosphate buffer solutions, respectively. In 1 M NaOH, the Faradaic efficiency for chiral and achiral catalysts is the same within error, e.g., a ratio of 1; however, as the pH decreases, the chiral catalysts become more efficient than their achiral analogs. This behavior is consistent with mechanisms proposed in previous reports for thin film chiral cobalt oxides[12] and has been attributed to increased generation of the $H_2O_2$ by-product for racemic catalysts, whereas the spin-polarized reaction intermediates on chiral catalysts promote the reaction pathway for triplet $O_2$.

## Spin effects on the reaction mechanism

To elucidate the origin of catalytic improvement from chirality, we examined the reaction mechanism for the OER catalysts by Tafel and reaction order analyses. As is common, we assume that the reaction proceeds through elementary steps involving single electron transfer events, albeit sometimes with a corresponding proton transfer. To be specific, consider the following mechanism that has been proposed for basic solutions without (a) and with (b) spin considerations[37]

| No Spin | Spin-polarized |
|---|---|
| (1a) M(III)-OH + OH$^-$ → M(IV)=O + H$_2$O + e$^-$ | (1b) M(III, ↓)-OH + OH$^-$ → M(III, ↓)-O(↑) + H$_2$O + e$^-$ |
| (2a) M(IV)=O + OH$^-$ → M(III)-OOH + e$^-$ | (2b) M(III, ↓)-O(↑) + OH$^-$ → M(III, ↓)-OOH(↑↑) + e$^-$ |
| (3a) M(III)-OOH + OH$^-$ → M(IV)=OO$^-$ + H$_2$O + e$^-$ | (3b) M(III, ↓)-OOH(↑↑) + OH$^-$ → M(III, ↓)-OO$^-$(↑↑) + H$_2$O |
| (4a) M(IV)=OO$^-$ → M(III)-OO | (4b) M(III, ↓)-OO(↑↑) → M(II, ↓) + O$_2$ + e$^-$ |
| (5a) M(III)-OO + OH$^-$ → M-OH+ O$_2$ + e$^-$ | (5b) M(II, ↓) + OH$^-$ → M(III, ↓)-OH + e$^-$ |

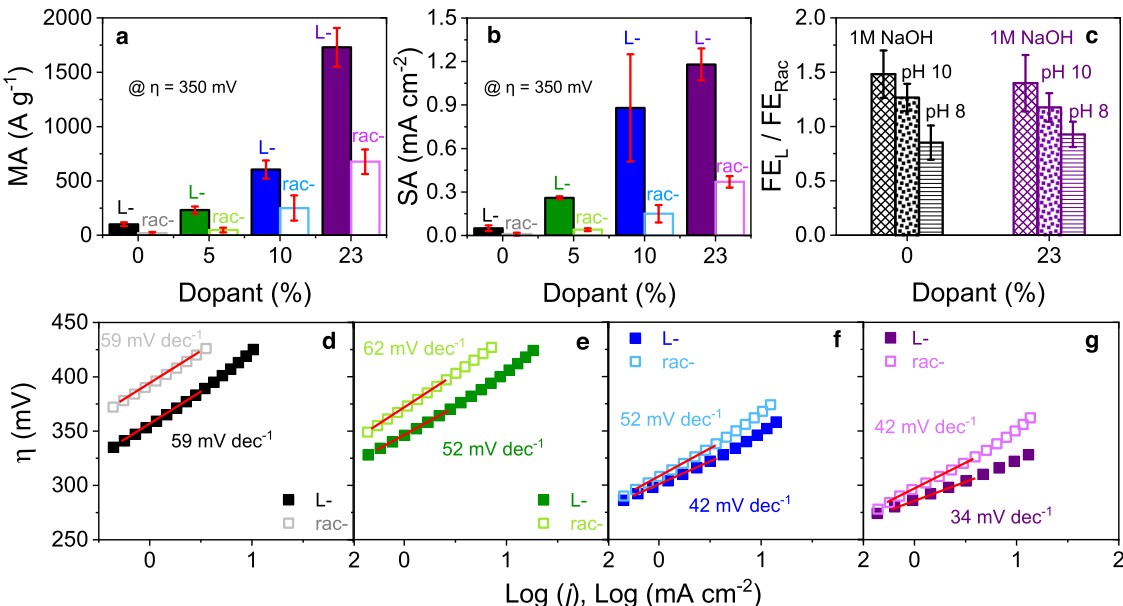

**Fig. 3 | Catalytic activity and characterization. a, b** Show mass activity, "MA" and specific activity "SA" of undoped (black) and 5% (green), 10% (blue), and 23% (purple) iron-doped Rac- (open) and L-cobalt oxide (filled) nanoparticle catalysts in Nafion measured in a 1 M NaOH electrolyte solution. **c** Shows the enhancement in Faradaic efficiency for undoped (black) and 23% Fe-doped (purple) chiral catalysts, compared to their achiral analogs in 1 M NaOH (horizontal dash) and in a 0.1 M pH 10 sodium carbonate (dotted) and 0.02 M pH 8 potassium phosphate (cross-hatched) buffer solutions. The error bars represent the standard deviation across at least three independent electrode preparations. **d–g** Show Tafel plots and corresponding slopes for undoped and 5, 10, and 23% iron-doped Rac- (open symbol) and L-cobalt oxide (filled symbol) nanoparticles, respectively.

Figure 3d–g show Tafel plots for undoped (black), 5% (green), 10% (blue), and 23% (purple) Fe-doped Rac- (open symbol) and L-cobalt oxide (filled symbol) in the low overpotential regime. The incorporation of iron dopants acts to decrease the Tafel slope in a manner consistent with other works[38]. In each case, the chiral catalysts exhibit the same, or shallower, Tafel slopes than the achiral catalysts of the same composition.

Because Tafel analysis alone does not accurately describe the reaction mechanism for the OER, a pH dependence was conducted for the undoped and 23% Fe-doped cobalt oxide catalysts to ascertain the reaction order (see Methods for details). The undoped cobalt oxide gave reaction orders of 1 and 1.2 for achiral and chiral catalysts, respectively. A microkinetic analysis shows that a reaction order of 1 and Tafel slope of ~59 mV dec$^{-1}$ coincides with reactions (1a) or (1b) in the scheme above being the rate-determining step (RDS)[39]. Tafel slopes of 42 mV dec$^{-1}$ (achiral) and 34 mV dec$^{-1}$ (chiral) and reaction orders of 2.0 (achiral) and 2.6 (chiral) are observed for 23% Fe-doped cobalt oxide; see Supplementary Fig. S4. The change in reaction order and Tafel slope indicates a change in RDS upon Fe-doping[39]. Considering the adsorbate type mechanism shown above, a microkinetic analysis indicates, that reaction (2a) is the RDS for the achiral catalyst[40,41]. Conversely, a Tafel slope of 30 mv dec$^{-1}$ and reaction order of 3 signifies a step further along the catalytic cycle; i.e., reaction (3a) or (3b) would be the RDS[40,41]. That is, the addition of OH$^-$ to generate the peroxyl species is no longer rate-limiting. Although the 23% Fe-doped chiral catalysts do not quite reach a reaction order of 3 and Tafel slope of 30 mv dec$^{-1}$, they are closer to reactions (3a) or (3b) being the RDS rather than (2a) or (2b). We note that the experimental data were analyzed at voltages slightly above that of the low overpotential region to avoid current contributions arising from catalyst redox states. Thus, the slight decrease in reaction order and increase in Tafel slope are expected and consistent with simulated data[40].

How does chirality lead to such a change in RDS? The mechanism discussed here shows one form which considers the effects of the electron spin alignment in the catalytic cycle and one that does not. Previous computational studies, which analyze this scheme, show that the transition state energies for different steps in the OER are strongly affected by spin alignment on the catalyst surface[18,42]. As shown in the spin-polarized mechanism, only spin-down (↓) electrons are injected into the electrocatalyst (because of CISS) and this generates spin-polarized surface intermediates (↑) on the chiral catalyst. Note that the generation of spin-aligned intermediates begins with step (1) in which an oxyl radical (M-O•) is formed rather than the closed shell oxo species (M=O). The key differences between catalysts arise from the first reaction step; chiral catalysts generate an M-O· reaction intermediate whereas achiral catalysts give rise to M=O reaction intermediates. The sequence of spin-down electron transfers from the hydroxide anion into the electrode leads necessarily to high spin multiplicity reaction intermediates on the catalyst surface, with subsequent generation of triplet oxygen. Conversely, no spin selectivity in the electron injection to the anode leads to spin-paired intermediates and thus a spin flip is necessitated to form the final triplet product[37]. Seemingly, the rate differences for progression through (2a) and (2b) for 23% iron-doped Rac-cobalt oxide and 23% iron-doped L-cobalt oxide are large enough that a change in RDS manifests. Note, such behavior is likely not the cause for the suppression of singlet-mediated byproducts, rather we attribute this phenomenon to spin restrictions on the reaction, not thermodynamic effects.

This work explores the pivotal role of electron spin polarization, generated through the chiral-induced spin selectivity effect, in facilitating the oxygen evolution reaction. We show that chiral Co$_{(3-x)}$Fe$_x$O$_4$ nanoparticle electrocatalysts decrease the overpotential for oxygen evolution beyond that found for their achiral counterparts, which already lie near the apex of most volcano plots. Thus, chirality acts to improve the overpotential beyond the simple constraints set forth by considering the adsorption energies of reaction intermediates. The electrical efficiency for chiral catalysts are >2-fold larger compared to their achiral equivalents and displays three to four times improvements in their mass activities and specific activities. The chiral Co$_{2.3}$Fe$_{0.7}$O$_4$ catalysts possess a >400-fold and >200-fold larger mass activity and specific activity, respectively, than benchmark IrO$_2$ catalysts. Moreover, the Faradaic efficiency studies show an improved reaction selectivity performance of chiral catalysts over their achiral analogs, and suppression of hydrogen peroxide byproducts, especially at low pH. Tafel analysis, in conjunction with reaction order studies, reveals that chirality changes the RDS. Thus spin alignment at chiral electrocatalyst surfaces is a viable strategy for improving the OER efficiencies of top-performing catalysts, and the ability of a catalyst to spin-polarize radical surface intermediates should be an important OER catalyst design criterion.

Reactions involving O$_2$, with its $^3\Sigma_g^-$ ground state, are broadly important in chemistry, biology, and the earth sciences, however, many other reactions proceed through radical intermediates as well. Thus, we expect that spin-polarized electron currents should affect the selectivity and efficiency of many reactions. While previous experimental demonstrations of spin selectivity in chemical reactions exist, they have used organized chiral structures on the surface of macroscopically planar electrodes whereas the results presented here show that chiral features on nanocatalysts with no net spatial direction in the laboratory frame display chiral selectivity. This finding implies that the nanoscale organization of chiral catalyst surfaces - be they chiral metal oxides, biological enzymes, or something else - display CISS and lead to spin selectivity in reaction pathways. By extension, chiral nanoparticle catalysts can thus be exploited for large surface area electrodes, towards much larger scale reactions, without loss or hindrance of the spin selectivity.

## Methods

### Synthesis of undoped chiral and racemic cobalt oxide
Undoped Co$_3$O$_4$ NPs were synthesized following a previously published literature strategy[29]. In this method, 15 mL of DI water, 2.5 mL of 100 mM NaBH$_4$, 2 mL of 100 mM L- or DL-cysteine, 2 mL of 100 mM sodium citrate, and 1 mL of 200 mM cobalt (II) chloride was added to a round-bottom flask and stirred for 2 h at ambient conditions; 23–25 °C. Following stirring, the solution turned an optically transparent dark brown color and indicated the successful synthesis of Co$_3$O$_4$ nanoparticles. The nanoparticles were then purified by precipitating the solution with the addition of a sevenfold larger volume of isopropanol followed by centrifugation for 20 min at ~9400 G. The nanoparticles were dried and then dispersed in water.

### Synthesis of Fe-doped chiral and racemic cobalt oxide
To synthesize Fe-doped cobalt oxide, iron (III) hexachloride, at 5, 10, and 23 mole percent, were added to the aforementioned cobalt chloride solution. As before, the synthesis of the chiral and racemic cobalt oxide was equivalent with the exception of the handedness of the ligand used during the synthesis. Purification of the doped materials was performed in the same manner as that described for the undoped materials.

### Characterization
Circular dichroism measurements were performed using a Jasco J810 CD spectrometer with an integration time of 4s and a bandwidth of 1 nm. UV-Vis absorbance measurements were made using an Agilent (model 8453) spectrometer. Electron microscopic images were acquired using a Thermo Scientific Titan Themis G2 200 probe Cs corrected transmission electron microscope (TEM) in scanning transmission electron microscopy mode (STEM) with a high-angle annular dark-field (HAADF) detector. It was operated at an acceleration voltage of 200 kV and a beam current of 100 pA. All HAADF STEM images were

recorded at 2048 × 2048 pixels. The TEM specimens were prepared from solutions of 5 mM of achiral or chiral nanoparticles in distilled H$_2$O; for each specimen, 10 μL of the solution was dropcast onto pure carbon film of ~20 nm supported by Au TEM grids. After ~5 min the excess solution was removed with a small piece of filter paper or lint-free clean-room cloth. Each TEM specimen was plasma cleaned using a Tergeo EM Plasma Cleaner before it was inserted into the TEM for examination. Inductively coupled plasma optical emission spectroscopy (ICP-OES) was performed using an argon flow with an Agilent 5100 VDV ICP-OES instrument. An aqua regia solution (3:1 ratio of hydrochloric acid to nitric acid) was prepared using ultrapure reagents (Sigma-Aldrich, HCl >99.999% trace metal basis; HNO$_3$ > 99.999% trace metal basis), a portion of which was then diluted with NANOpure water to yield a 5% v/v aqua regia mixture. An aliquot of the catalyst solution was dissolved overnight in ~100 μL of the concentrated aqua regia and then diluted to ~10 mL with the 5% aqua regia and analyzed via ICP-OES to determine the catalyst stoichiometry. The metal concentrations were determined by comparison to a seven-point standardization curve with a range of 0.10–10 ppm for each metal (0.10, 0.50, 1.0, 2.5, 5.0, 7.5, and 10 ppm) and was prepared by volume using ICP standards (Fluka, TraceCERT 1000 ± 2 mg/L metal in HNO$_3$) diluted in a 5% aqua regia matrix. A 3 min flush time with a 5% nitric acid matrix was used between all runs, and a blank was analyzed before each unknown sample to confirm the removal of all residual metals from the instrument.

## Electrochemical measurements

Electrochemical measurements were performed using a glassy-carbon (GC) working electrode (geometrical area = 0.07 cm$^2$) mechanically polished using 0.05-micron-sized alumina slurry to a mirror finish. A catalyst solution was prepared by mixing 0.5 mg of synthesized nanomaterial catalyst with 12.5 μL of Nafion perfluorinated resin solution (Aldrich) and 250 μL water/isopropyl alcohol (3:1 v/v). After homogeneous dispersion by sonication for 10 min, a 1 μL aliquot was dropcast onto the GC electrode. The electrode was then dried at 70 °C for 30 min to evaporate the solvents, leading to a catalyst loading of 0.027 mg cm$^{-2}$. The electrochemical experiments were carried out using a 618B CH Instruments potentiostat in a 1 M NaOH electrolyte solution, using an Ag/AgCl reference electrode (CHI) and Pt wire as the counter electrode, unless otherwise specified. Linear sweep voltammetry (LSV) experiments were collected at a scan rate of 10 mV s$^{-1}$. The electrochemical results reported in this work were *iR* compensated using the CH instruments software package command prior to measurements. Measurements reported in units of RHE were converted using the equation $E_{RHE} = E_{Ag|AgCl} + E^0_{Ag|AgCl} + 0.059 \times pH$, where $E_{Ag|AgCl}$ is the potential vs Ag|AgCl reference electrode and $E^0_{Ag|AgCl} = 0.197$ V.

For the determination of electrochemical surface area, ECSA, cyclic voltammograms were taken in the non-Faradaic region (0.45 to +0.5 V vs. Ag|AgCl reference electrode) without stirring the solutions at scan rates from 10 to 70 mV s$^{-1}$. ECSA was calculated using the formula; ECSA = $C_{dl}/C_s$ where $C_{dl}$ is the double layer capacitance obtained from the slope of the current (charging current at a constant potential) vs. scan rate plot and $C_s$ is the specific capacitance of the material. A specific capacitance value of 40 μF cm$^{-2}$ was used to calculate the ECSA of nanomaterials[43]. While $C_s$ value for the doped catalysts likely deviates from that of the undoped catalysts, the chiral and achiral catalysts are the same. Mass activities and specific activities of all of the catalysts were determined using the equations: Mass activity = $j/m$, Specific activity = $j/RF$, where m, $j$, and RF refer to mass loading, the current density at 350 mV, and roughness factor, respectively. Note, RF = ECSA/Geometric area of the electrode

Faradaic efficiency measurements were performed using a rotating ring-disk electrode, RRDE, apparatus from ALS Co. RRDE-3A, and a CH Instruments 750c bipotentiostat. A glassy carbon disk (geometric area = 0.12 cm$^2$) and platinum ring (ALS) were used as

the working electrodes. Prior to each catalyst loading the electrodes were polished using 0.05-micron-sized alumina slurry followed by sonication for 30 s in H$_2$O and dried under an Ar stream. Platinum wire was used as the counter electrode and Ag|AgCl (ALS RE-1B) was used as the reference electrode. The catalyst was prepared in the same manner as that described above, using a dropcast method, and was applied only to the disk electrode. The electrolyte was purged with Ar for 30 min prior to RRDE measurements. To ensure a fresh Pt surface at the ring electrode, which was clear of any catalyst deposition, the ring was cycled between −0.03 and 1.37 V vs RHE at a scan rate of 500 mV s$^{-1}$ for 50 cycles. Linear sweep voltammetry measurements were performed at a 10 mV s$^{-1}$ scan rate and a rotation rate of 1600 rpm. The ring was held at a constant potential of 0.1 V vs RHE so that the oxygen produced at the disk was reduced at the ring. Measurements were performed in 1 M NaOH, in a 0.1 M sodium carbonate/bicarbonate buffer solution (pH 10), and in 0.02 M phosphate buffer (pH 8). The improvement in Faradaic Efficiency was calculated using the equation below at 2 mA/cm$^2$ current density defined the by geometric area;

$$\frac{FE_L}{FE_{Rac}} = \frac{\left[\frac{I_{ring}}{I_{disk}}\right]_L}{\left[\frac{I_{ring}}{I_{disk}}\right]_{Rac}}$$

where $FE_L$ and $FE_{Rac}$ is the Faradaic efficiency of the chiral and racemic electrode materials, respectively, and $I_{ring}$ and $I_{disk}$ are the current densities of the ring and disk electrodes, respectively, at 2 mA/cm$^2$. Each measurement was repeated for at least three independently prepared catalyst electrodes.

To determine the reaction order of the undoped and 23% Fe-doped catalyst materials, NaOH solutions with varying pH values of 13.1, 13.3, and 13.6 were prepared. In order to maintain the ionic strength of the solution, KNO$_3$ was used as an additive in the dilute NaOH solutions. The reaction order was then estimated from the slope of a log $i$ (mA/cm$^2$) vs. log [OH$^-$] plot at a given overpotential. Note, different potentials had to be chosen for the different catalysts because of the changes in overpotential, thus the linear region of the Tafel curves at low overpotential were used.

## Data availability

All data needed to evaluate and reproduce the conclusions is provided in the manuscript or supplemental information. Additional information is available upon request. Source data are provided with this paper.

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

## Acknowledgements

The authors acknowledge financial support from the NSF–DFG project "Echem: Spin-polarized electrons for spin selective electrocatalysis" by the United States National Science Foundation (NSF) CHE- 2140249 (D.H.W.) and Professor J. E. Millstone for helpful advice.

## Author contributions

B.P.B. and D.H.W. conceptualized the project. A.V. synthesized the NPs and performed the electrochemical experiments. C.C. performed the RRDE experiments. S.T. collected the TEM images and K.N.K. measured the ICP-OES. A.V., B.P.B., C.C., and K.N.K. analyzed the data. B.P.B. wrote the initial draft and all of the authors contributed to the review and editing. D.H.W. was responsible for the project funding acquisition and administration.

## Competing interests

The authors declare no competing interests.
