## [Peer Review File · Nature Communications]

Chiral electrocatalysts eclipse water splitting metrics through spin controlREVIEWER COMMENTS

Reviewer #1 (Remarks to the Author):

The manuscript by Vadakkayil et al, demonstrates that novel chiral nanoparticle-catalysts display a 400-fold higher mass activity compared to benchmark IrO₂ catalysts. The work is original and the results are outstanding. Hence, the work should be published in Nature Communications.

However, I have one comment:

The authors found a change in the rate determining step by chirality and it was explained by suppression of singlet-mediated byproduct. The explanation is persuasive, however, is there any evidence that support the suppression of singlet-mediated byproduct? Did the authors confirm the suppression of singlet O₂ or H₂O₂ formation?

Before publication, the authors are requested to address the above comment.

Reviewer #2 (Remarks to the Author):

The manuscript entitled "Chiral electrocatalysts eclipse water splitting metrics through spin control" by Aravind Vadakkayil et al. demonstrates an enhancement in the water splitting reactions, due to a synergetic effect between an improvement in mass transport and the specific electrochemical activity. This was achieved by exploiting chiral Fe-doped cobalt oxide nanoparticles as catalysts. The authors claimed that chirality plays a crucial role in the electrocatalytic processes, affecting the reaction pathways through an electron spin effect. Moreover, the overpotential can be decreased by suppressing the formation of some byproducts.

Since in the last decade the CISS effect is increasingly used for different applications, I consider that this work presents significant results, not only in the field of heterogeneous catalysis, but even in the one of spintronic.

Conclusions are well supported by the presented results and the figures are clear and intuitive. The methodology is well described as well as detailed enough to allow the work to be reproduced.

I recommend this paper to be published in Nature Communications after taking in consideration some questions/curiosities for the Authors:

- i) Do the Authors have considered the use of other paramagnetic metals respect to cobalt and iron? Since Co is not an environmentally friendly material.
- ii) Do the Authors already plan how to scale-up the process?
- iii) Why for the ECSA calculation the Authors do not use impedance spectroscopy?
- iv) Can the Authors provide some explanations to the considerable high error bar, in comparison to the others, for the blue column in Figure 3b?
- v) How does the Authors know that they are depositing always the same quantity of catalyst on the GC electrode?
- vi) The Authors changed the pH using different (buffer) solutions. Do they try to change just the ionic strength? Have they considered to keep constant the nature of the ions in the solutions (I mean not passing from NaOH, sodium carbonate/bicarbonate and phosphate buffer) just by adjusting the pH?
- vii) I feel that the chirality effect is not completely exploited, since the electrons can be considered chiral due to their rototranslational motion. Maybe in the explanation of the enhancement of the mass transport and specific activity for the water splitting, the Authors can consider a synergetic effect between different "types" of chirality (molecular chirality of nanoparticles and chirality of the electrons during their motion).
- viii) Concerning mass transport, when the Authors refer to mass enhancement, are they talking about mass transport?
- ix) For my curiosity: in the case of mass transport, what regime is it (if it is not diffusion)? Is it convection or magnetohydrodynamic convection?

RESPONSE TO REVIEWERS' COMMENTS

We would like to thank the reviewers for taking the time and energy to review the manuscript and appreciate the valuable feedback to help improve the quality of our work. In the following, we provide a point-by-point response to the reviewer's comments.

Reviewer 1

The manuscript by Vadakkayil et al, demonstrates that novel chiral nanoparticle-catalysts display a 400-fold higher mass activity compared to benchmark IrO₂ catalysts. The work is original and the results are outstanding. Hence, the work should be published in Nature Communications.

We thank the reviewer for the positive comment.

The authors found a change in the rate determining step by chirality and it was explained by suppression of singlet-mediated byproduct. The explanation is persuasive, however, is there any evidence that support the suppression of singlet-mediated byproduct? Did the authors confirm the suppression of singlet O₂ or H₂O₂ formation?

We attribute the change in rate determining step to the chirality induced spin alignment on the catalysts. Using the microkinetic analysis given, the chiral catalysts proceed through step 2 of the reaction mechanism (generation of a spin polarized peroxy species) more readily than do the achiral catalysts. A similar argument has been proposed for magnetic catalysts in which the transition state energies of the reaction intermediates change upon application of an external magnetic field (see *Curr. Opin. Electrochem.* **30**, 100804, (2021) and Nature Communications **volume 12**, Article number: 2608 (2021). While this can suppress singlet mediated byproducts as well, we believe that spin constraints on the singlet reaction proceeding is more influential; e.g. spin alignment gives a spin-forbidden reaction step for a singlet reaction pathway. Indeed, the Faradaic efficiency (Figure 3c) shows that peroxide formation is inhibited for the chiral catalysts over achiral catalysts.

To clarify this point, we have added the following sentence to the document
“Note, such behavior is likely not the cause for the suppression of singlet-mediated byproducts, rather we attribute this phenomenon to spin restrictions on the reaction, not thermodynamic effects.”

Reviewer 2

The manuscript entitled “Chiral electrocatalysts eclipse water splitting metrics through spin control” by Aravind Vadakkayil et al. demonstrates an enhancement in the water splitting reactions, due to a synergetic effect between an improvement in mass transport and the specific electrochemical activity. This was achieved by exploiting chiral Fe-doped cobalt oxide nanoparticles as catalysts. The authors claimed that chirality plays a crucial role in the electrocatalytic processes, affecting the reaction pathways through an electron spin effect. Moreover, the overpotential can be decreased by suppressing the formation of some byproducts. Since in the last decade the CISS effect is increasingly used for different applications, I consider that this work presents significant results, not only in the field of heterogeneous catalysis, but even in the one of spintronic. Conclusions are well supported by the

presented results and the figures are clear and intuitive. The methodology is well described as well as detailed enough to allow the work to be reproduced.

We thank the reviewer for the supportive comments.

I recommend this paper to be published in Nature Communications after taking in consideration some questions/curiosities for the Authors:

1.) Do the Authors have considered the use of other paramagnetic metals respect to cobalt and iron? Since Co is not an environmentally friendly material.

While cobalt mining is not the most environmentally friendly process, Co is a comparatively low-cost earth abundant transition metal and is viewed as an attractive alternative to precious metal catalysts such as ruthenium and iridium oxides. Recent work on cobalt oxides and hydroxide catalysts are now considered to be among the best performing catalyst systems for OER; See *Nat. Energy*. (2022) doi.org/10.1038/s41560-022-01083-w, *Angew. Chem. Int. Ed.* **54**, 8722-8727, (2015). Indeed, the CISS effect has been shown to improve the catalytic properties of other materials, such as iron oxide and copper oxide (*J. Phys. Chem. C* 2019, 123, 3024–3031), however their catalytic performance pails in comparison to that which is reported here. One of the important findings from this study was to demonstrate that symmetry constraints can improve the Faradaic efficiency and overpotential of catalysts that are already considered state-of-the-art materials from thermodynamic considerations (i.e. traditional volcano plots).

2.) Do the Authors already plan how to scale-up the process?

This is an excellent question and something that the findings reported in the manuscript support. Our findings indicate that the chirality effects on the OER are localized on the catalyst surface, and as such, can be extrapolated to much larger areas without loss or hindrance to the spin selectivity. Therefore, other catalyst supports, such as Ni foam, can conceivably be employed to scale the process.

We have added the following sentence to the conclusion of the manuscript to hopefully better clarify this point

“By extension chiral nanoparticle catalysts can thus be exploited for large surface area electrodes, towards much larger scale reactions, without loss or hindrance of the spin selectivity.”

3.) Why for the ECSA calculation the Authors do not use impedance spectroscopy?

While it's true that EIS can also be used to calculate the ECSA. It has become normal practice to use double layer capacitance measurements to determine ECSA and these have been found to be in good agreement with EIS measurements, see *J. Am. Chem. Soc.* 2013, 135, 16977–16987 (the difference in double layer capacitance calculated by both methods are within $\pm 15\%$ error).

4.) Can the Authors provide some explanations to the considerable high error bar, in comparison to the others, for the blue column in Figure 3b?

Figure 3b shows the specific activity of the nanomaterials, calculated by dividing the current at 350mV potential by the ECSA of the material. We prepared 3 independent electrodes for each nanomaterial and for this particular dataset there was some additional variability. Despite the large error compared to the rest of the data in the plot it is important to note that the specific activity of 10%-L is significantly different from that of the racemic analogue. Moreover, the trend of the chiral nanomaterial exhibiting 2-3 times higher specific activity than its racemic analogue holds true in this case as well.

5.) How does the Authors know that they are depositing always the same quantity of catalyst on the GC electrode?

This is a very important question because reproducibility is paramount for making comparisons across different catalytic systems. The deposition of the catalyst on the electrode surface is performed by taking 0.5mg of each catalyst material and mixing it with a constant amount of water, IPA and Nafion. The mixture was then sonicated to a homogeneous ink and 1 μ L of this solution was dropcast onto the GC electrode. Therefore, the amount of catalyst material should be the same among different preparations. That being said, the process of drying is difficult to control and can lead to differences in aggregation / electrochemical surface area for each electrode. To account for these differences, every measurement was made in triplicate (three separately prepared electrodes) and standardized performance metrics, such as specific activity, which can account for slight deviations in ECSA, are included in the analysis. Lastly, the good reproducibility among measurements (relatively small error bars compared to the difference between chiral and achiral response) indicate that any of the observed effects associated with chirality are much larger than differences that can arise from electrode/catalyst preparation.

6.) The Authors changed the pH using different (buffer) solutions. Do they try to change just the ionic strength? Have they considered to keep constant the nature of the ions in the solutions (I mean not passing from NaOH, sodium carbonate/bicarbonate and phosphate buffer) just by adjusting the pH?

The above mentioned 3 different electrolytes were made for comparing the Faradaic efficiency of chiral catalysts over their achiral analogues. Because decreasing pH is known to increase the favorability for H₂O₂ formation (J. Phys. Chem. C 2020, 124, 22610–22618) we chose to investigate how chirality can be used to impede this process. While the reviewer is correct that solution conditions, such as ionic strength, can also influence the Faradaic efficiency the focus of this particular series of experiments was to determine the differences between chiral and achiral, not optimize the conditions to maximize the efficiency.

For the reaction order experiments the ionic strength was kept constant by adding KNO₃ salt to NaOH solutions prepared at varying pH values. These experiments were performed in accordance with previously published methods (see Int. J. Hydrog. Energy, 33, 4936-4944, (2008)) and are necessary for accurately determining the reaction order.

7.) I feel that the chirality effect is not completely exploited, since the electrons can be considered chiral due to their roto-translational motion. Maybe in the explanation of the enhancement of the mass transport and specific activity for the water splitting, the Authors can consider a synergetic effect between different “types” of chirality (molecular chirality of nanoparticles and chirality of the electrons during their motion).

In this work the chiral induced spin selectivity effect is responsible for electron spin polarization in the ‘chiral’ catalysts outperforming their ‘achiral’ analogues in which spin polarization does not occur. The ‘molecular chirality’, with an enantiopure ligand shell imprints chirality onto the catalyst’s electronic states and gives rise to the spin polarized electrons, however it must be noted that the control experiments use a racemic mixture (L- and D-cysteine) and so we do not believe the ligands themselves are contributing to the OER differently, more so the spin selectivity which manifests from the chiral imprinting. Therefore any ‘molecular chirality’ effects should be accounted for and only the electron spin effects are being studied. Thus, this behavior is analogous to other studies performed on ferromagnetic materials in which application of an external magnetic field spin polarizes the electrons and improves the OER characteristics; See Nature Energy 4, 519–525 (2019).

8.) Concerning mass transport, when the Authors refer to mass enhancement, are they talking about mass transport?

It appears that the reviewer may be confusing mass transport with mass activity; mass activity = j/m where j , m , refer to current density at 350 mV, catalyst mass loading respectively. This is a common metric in the catalysis community to assess how catalyst loading affects performance.

9.) For my curiosity: in the case of mass transport, what regime is it (if it is not diffusion)? Is it convection or magnetohydrodynamic convection?

As described in the previous question, mass transport is not an important concern; as the reactant is water (nominally 55 M). Also, we note that there is no magnetic field applied so magnetohydrodynamic effects are not present.

REVIEWERS' COMMENTS

Reviewer #1 (Remarks to the Author):

I was satisfied with the author's responses.
Now I can recommend the publication in Nature Communications.

Reviewer #2 (Remarks to the Author):

The authors have fully answered all of my concerns and therefore I recommend publishing as it is.